# Impact of Straw Incorporation on the Physicochemical Profile and Fungal Ecology of Saline–Alkaline Soil

**DOI:** 10.3390/microorganisms12020277

**Published:** 2024-01-28

**Authors:** Weiming Ma, Li Ma, Jintang Jiao, Abbas Muhammad Fahim, Junyan Wu, Xiaolei Tao, Yintao Lian, Rong Li, Yapeng Li, Gang Yang, Lijun Liu, Yuanyuan Pu, Wancang Sun, Wangtian Wang

**Affiliations:** 1State Key Laboratory of Aridland Crop Science, Gansu Agricultural University, Lanzhou 730070, China; 18409442030@163.com (W.M.); 13679302671@163.com (J.J.); fahimabbaskhan@yahoo.com (A.M.F.); txl162185@163.com (X.T.); lianyintao@163.com (Y.L.); 17834453340@163.com (R.L.); 15379222259@163.com (Y.L.); yangang1018@163.com (G.Y.); liulj198910@163.com (L.L.); vampirepyy@126.com (Y.P.); sunwanc@gsau.edu.cn (W.S.); wangw@gsau.edu.cn (W.W.); 2College of Agronomy, Gansu Agricultural University, Lanzhou 730070, China

**Keywords:** saline–alkali soil, straw returning, environmental factors, microbial communities, network analysis

## Abstract

Improving the soil structure and fertility of saline–alkali land is a major issue in establishing a sustainable agro-ecosystem. To explore the potential of different straw returning in improving saline–alkaline land, we utilized native saline–alkaline soil (SCK), wheat straw-returned saline–alkaline soil (SXM) and rapeseed straw-returned saline–alkaline soil (SYC) as our research objects. Soil physicochemical properties, fungal community structure and diversity of saline–alkaline soils were investigated in different treatments at 0–10 cm, 10–20 cm and 20–30 cm soil depths. The results showed that SXM and SYC reduced soil pH and total salinity but increased soil organic matter, alkali-hydrolyzable nitrogen, available phosphorus, total potassium, etc., and the enhancement effect of SYC was more significant. The total salinity of the 0–10 cm SCK soil layer was much higher than that of the 10–30 cm soil layers. Fungal diversity and abundance were similar in different soil layers in the same treatment. SXM and SYC soil had higher fungal diversity and abundance than SCK. At the genus level, *Plectosphaerella*, *Mortierella* and *Ascomycota* were the dominant groups of fungal communities in SXM and SYC. The fungal diversity and abundance in SXM and SYC soils were higher than in SCK soils. Correlation network analysis of fungal communities with environmental factors showed that organic matter, alkali-hydrolyzable nitrogen and available phosphorus were the main environmental factors for the structural composition of fungal communities of *Mortierella*, *Typhula*, *Wickerhamomyces*, *Trichosporon* and *Candida*. In summary, straw returning to the field played an effective role in improving saline–alkaline land, improving soil fertility, affecting the structure and diversity of the fungal community and changing the interactions between microorganisms.

## 1. Introduction

Soil salinization is a major environmental and socioeconomic issue worldwide, and climate change is likely to worsen it. It limits agriculture production and regional food production. Saline–alkali soil covers 9.5 × 10^8^ hm^2^ globally, including 9.9 × 10^8^ hm^2^ in China [1], primarily in Northwest China, North China, Northeast China and Southeast Coast regions, with a 16.67% agricultural improvement potential [2]. In arid climates with low precipitation, the dominant upward movement surpasses the downward movement, leading to the accumulation of salinity in the uppermost layer of soil [3]. Saline–alkali soil is also a valuable cultivated land resource that holds significant potential for utilization. Consequently, the levels of soluble salts and cations exceed the typical range, rendering the soil inhospitable for the normal growth of the majority of plant species [4]. The advancement of cost-effective, efficient and environmentally friendly technologies for improving and utilizing saline–alkali soil, as well as the widespread adoption of comprehensive utilization practices, can serve as a robust safeguard for global food security [5]. Currently, the main strategies for enhancing saline–alkali soil primarily revolve around physical, chemical, engineering and biological improvement measures [6,7,8,9,10,11,12]. However, it should be noted that physical, chemical and engineering practices to improve soil may result in agricultural pollution, soil structure deterioration, as well as increased expenses [13]. Biological improvement measures have significant benefits and are sustainable.

The utilization of straw as a natural organic fertilizer, which can not only avoid the air pollution caused by incineration but, more importantly, can cultivate soil fertility, improve soil structure, reduce soil salinity, increase porosity, promote microbial vitality, the development of crop roots and protect the environment, etc., plays an important role [14,15,16]. In recent years, there has been a significant amount of research conducted on the enhancements in saline–alkali soil through the practice of straw incorporation. It has been observed that straw incorporation effectively reduces salinization indicators, such as soil bulk density, salinity content and pH value, in saline–alkali soil [17,18]. Additionally, a certain positive impact on crop yields has been observed [19]. Consequently, the incorporation of straw into the soil has emerged as a crucial strategy for improving soil quality. The investigation of straw incorporation as a means of treating saline–alkali soil holds substantial importance for the development and utilization of cultivated land resources.

Furthermore, microorganisms plays a crucial role in the soil ecosystem and exhibit high sensitivity towards saline–alkali-induced ion stress [20]. As soil salinity increases, the susceptible microbial populations gradually decline, while strains capable of withstanding high-salinity concentrations progressively dominate. Consequently, the proliferation of salinity-intolerant fungi is restricted [21]. Studies have shown that the correlations among soil chemical indicators were enhanced after straw addition in saline soils, and fungal decomposers have common and specific responses in soil [22]; straw returning increases the diversity of soil bacteria and fungi, promotes soil microbial nitrogen cycling, provides nitrogen to soil [23] and improves soil-active organic carbon composition, enzyme activity and microbial ecological function [24]. The fluctuation pattern of physical and chemical attributes, such as soil moisture content, organic matter, pH and available potassium, exhibits a strong correlation with the abundance and composition of microorganisms inhabiting the soil. This, in turn, influences the overall structure of the soil microbial community [25,26,27]. Various environmental factors can impact soil microorganisms, with a primary emphasis on the physical and chemical properties of the soil. These properties serve as highly sensitive biological indicators, capable of predicting alterations in soil nutrients and overall soil health [28].

The arid region located in Northwest China exhibits a typical temperate continental arid climate. This climate is characterized by significant soil evaporation, a high concentration of soil parent matter, limited freshwater resources and a heavy reliance on irrigation for agricultural land. Consequently, agricultural production suffers substantial losses. Therefore, it is imperative to address and enhance the management of soil salinization. In this study, we employed fungal microbial community structure characteristics, redundancy analysis (RDA) and correlation network analysis to investigate the relationship between soil microbial community structure and environmental factors in saline–alkali soil under various straw-returning treatments. Overall, our study employed diverse microbial community analysis techniques to demonstrate the complex interplay between straw incorporation, environmental factors and soil microbial communities in saline–alkali soil. Notably, straw treatment significantly influenced the abundance and diversity of specific microbial groups, suggesting their potential as bioindicators for optimizing salinity management strategies. These findings pave the way for developing more targeted and efficient straw incorporation techniques, specifically tailored for Northwest China’s arid climate and soil conditions. By harnessing the power of soil microbial communities, we can unlock promising avenues for enhancing agricultural productivity, promoting sustainable land management and, ultimately, contributing to greater food security throughout the region.

## 2. Results

### 2.1. Effects of Different Treatments on pH, Total Salinity and Nutrients in Saline Soils

#### 2.1.1. Effects of Different Treatments on Soil pH and Total Salinity of Saline–alkali Soil

The application of wheat straw (SXM) and rape straw (SYC) to saline–alkali soil demonstrated notable effects on soil pH and total salinity. Both straw treatments resulted in reductions in soil pH and total salinity, with a more pronounced impact on salinity, as illustrated in Figure 1A,B. When examining depth-specific effects, the control group (SCK, no straw) displayed relatively consistent soil pH across depths (0–10 cm, 10–20 cm, 20–30 cm). However, total salinity exhibited a significant increase in the top 0–10 cm layer (SCK1) compared to the deeper layers (SCK2 and SCK3), indicating severe surface salinization (Figure 1B). Interestingly, there were no significant differences observed between wheat and rape straw layers concerning their effects on pH and salinity. The incorporation of both types of straw yielded similar outcomes, suggesting comparable efficacy in mitigating soil salinity and pH levels.

#### 2.1.2. Effects of Different Straw Returning on Soil Nutrients in Saline–Alkali Soil

Applying straw (SXM and SYC) significantly increased organic matter (OM), available nitrogen (AN), available phosphorus (AP), available potassium (AK), total nitrogen (TN), total phosphorus (TP) and total potassium (TK) in saline–alkali soil compared to the unamended control (SCK). These increases were diverse across soil depths (illustrated in Figure 1C–I). For SXM, the 10–20 cm layer had the highest levels of AN, AP, TN and TK, while the 20–30 cm layer showed elevated TP and AK. The 0–20 cm layer exhibited increased OM. In contrast, SYC’s 10–20 cm layer had the highest concentrations of AP, TN, TP and TK, while the 0–10 cm layer displayed the most AK.

### 2.2. Distribution of OTUs and Community Diversity of Soil Fungi under Different Treatments

#### 2.2.1. Distribution of OTU Number in Soil Fungi under Different Treatments

The findings indicated that fungal operational taxonomic units (OTUs) were identified in the soil layers of unimproved saline–alkali soil (SCK), as well as in saline–alkali soil that had been modified using wheat straw (SXM) and rapeseed straw (SYC). In contrast, the improved saline–alkali soil group (SXM and SYC) exhibited a significantly higher number of fungal OTUs compared to the saline–alkali soil control group (SCK). The number of fungal OTUs detected in the samples from the wheat straw-improved saline–alkali soil and rapeseed straw-improved saline–alkali soil accounted for 42.58% and 51.20% of the total fungal OTUs, respectively. Nevertheless, the proportion of fungal OTUs observed in the unimproved saline–alkali soil was merely 6.23%. (Appendix A).

To investigate the presence of identifiable common core microbiomes [28], Venn diagrams were utilized to compare samples subjected to different treatments. The Venn plots displayed overlapping regions, representing shared microbial taxa among the samples (Appendix A). Our analysis showed that a total of 8669 OTUs were detected, of which 724, 4567 and 5413 OTUs were identified in the SCK, SXM and SYC groups, respectively. There were a total of 1579 OTUs in the SXM and SYC groups, 41 OTUs in the SCK and SXM groups, 79 OTUs in the SCK and SYC groups and 168 OTUs between the three groups. The Venn diagram shows that there are unique OTUs in each group. Specifically, there are 436 OTUs in the SCK group, 2779 OTUs in the SXM group and 3587 OTUs in the SYC group. Consequently, SXM and SYC fungal communities were more numerous and diverse compared to SCK, which was associated with better soil health.

#### 2.2.2. Diversity of Soil Fungal Communities under Different Treatments

Alpha diversity analysis, using ACE, Simpson, Shannon and PD_whole_tree indices, revealed consistently higher fungal diversity in both straw-treated groups (SXM and SYC) compared to the untreated control (SCK) (Appendix A). Notably, SYC exhibited the highest diversity scores across all indices, with significant differences from SCK in all cases. However, SXM and SYC did not differ significantly in terms of PD_whole_tree.

To further visualize community structure, we performed principal component analysis (PCA), which revealed distinct clustering patterns (Appendix A). The first two principal components explained 94.68% of the variation, with SYC samples showing significant overlap, suggesting similarity in species composition. These results demonstrate that straw application, particularly rapeseed straw, significantly enhances fungal diversity in saline–alkali soil. The PCA results further highlight the distinct community structures associated with different treatments. To analyze whether the differences between sample points in different subgroups were significant, we used PERMANOVA for analysis (Appendix A). The results showed that the grouping differences were larger for SXM and SYC compared to SCK. The differences between groups were greater than those within groups when comparing within and between groups. This suggests that straw application led to an increase in soil fungal community species, making the soil microbial environment more complex.

### 2.3. Prediction of Soil Fungal Community Composition and Function under Different Treatments

The stacked plot (Figure 2A) reveals the dominant fungal genera at the topsoil level. Among the top 10, Plectosphaerella and Enterocarpus exhibited the most distinct responses to straw application. Plectosphaerella abundance significantly increased in SXM and SYC compared to the control (SCK), suggesting its potential role in enhanced soil health under these treatments. Conversely, Enterocarpus abundance sharply declined in SXM and SYC, while remaining high in SCK. This suggests possible beneficial effects of straw application in modifying the fungal community. Furthermore, it was observed that the relative abundance of Mortierella, Botryotrichum, Filobasidium, Ascomycota, Fusarium and Alternaria in the (SXM) and (SYC) treatments was greater compared to the (SCK) treatment alone. This suggests that the practice of straw returning facilitated the proliferation of these particular species.

The FunGuild fungal function prediction analysis was conducted using the greengenes database, as depicted in Figure 2B. Based on the classification and analysis of fungal communities using microecological guilds, it was observed that three distinct microecological guilds, namely pathotroph, symbiotroph and saprotroph, were identified across three distinct treatments and varying depths. There were no statistically significant differences observed in the relative abundance of the three main microbiota guilds at different depths (0–10 cm, 10–20 cm and 20–30 cm) within each treatment group. However, it was found that the relative abundance of soil phototrophic fungi in the SCK group (4.91%) was significantly lower compared to the SXM and SYC groups (40.42% and 32.79%, respectively). Additionally, the relative abundances of soil symbiotic fungi in the SCK, SXM and SYC groups were 2.40%, 15.20% and 9.68%, respectively Notably, the relative abundance of soil saprophytic fungi in the SCK group (92.68%) was significantly higher than that in the SXM and SYC groups (44.37% and 58.24%, respectively). Straw application elevated the relative abundance of pathogenic and symbiotic fungi. This indicates that these altered guild abundances suggest potential shifts in nutrient cycling and microbial interactions within the soil ecosystem.

### 2.4. Correlation Network Analysis of Soil Fungal Communities

To investigate the probable mechanisms of soil fungal communities under various treatment modalities, we conducted a comparative analysis of the co-occurrence networks among fungal taxa (Figure 3) The Spearman correlation coefficient is employed to quantify the proximity between different groups at the genus level. In comparison to the SCK group (*n* = 29), both the SXM group (*n* = 58) and the SYC group (*n* = 60) exhibited a higher number of genus-level fungal communities engaged in network interactions. However, as the number of fungal communities increased, the density of correlation networks decreased. Specifically, the network density values for the SCK, SXM and SYC groups were 0.0985, 0.0605 and 0.0565, respectively. In terms of community correlation at the genus level, the logarithmic positive correlation values between the communities in the SXM and SYC groups were 55 and 79, respectively. In contrast, the logarithmic positive correlation value between the communities in the SCK group was only 19.

### 2.5. Relationship between Soil Fungal Communities and Environmental Factors

This study employed the RDA method to investigate the impact of soil environmental factors on the composition of the top-10 fungal communities at the genus level (Figure 4). The findings revealed that soil pH, organic matter and total nitrogen exhibited the strongest associations, suggesting that these factors were more effective in elucidating the variations in relative abundance among the top 10-fungal species. Specifically, soil pH demonstrated a significant correlation with Ascomycota and Alternaria, while organic matter and total nitrogen exhibited significant correlations with Mortierella. The findings of this study indicated that the significance of soil environmental factors on the abundance of fungi followed the order of total nitrogen, organic matter, pH, alkali-hydrolyzable nitrogen, available phosphorus, total potassium, total salinity, available potassium and total phosphorus (Table 1). Among these factors, total nitrogen, organic matter, pH, alkali-hydrolyzable nitrogen and total salinity exhibited statistically significant effects on the abundance of fungi (*p* < 0.05). The percentages of the explanations for the disparities in alkali-hydrolyzable nitrogen and total salt content on the fungal level were found to be 47.74%, 42.78%, 41.66% and 31.68%, respectively. Conversely, the impacts of other factors on the fungal level were determined to be statistically insignificant (*p* > 0.05).

### 2.6. Network Analysis of the Correlation between Soil Fungal Communities and Environmental Factors

The findings of the study indicate a negative association between the pH and TS of the heatmap and several environmental parameters, while a positive correlation was observed with OM, AN, AP, AK, TP and TN (Pearson’s r > 0) (Figure 5). Notably, OM exhibits the strongest correlation with AP and AN, as well as AP alone, with Pearson’s r exceeding 0.5. A significant correlation (Mantel’s *p* < 0.01) was observed between the fungal community at the genus level and environmental factors. Specifically, the environmental factors OM, AN and AP had the strongest influence on the structure of the soil fungal community at the genus level (Mantel’s r ≥ 0.4). This suggests that OM, AN and AP play a crucial role in shaping the composition of the soil fungal community. Additionally, the Simpson index showed a correlation with AP and TP in the Alpha index, with a significance level of 0.01< Mantel’s *p* < 0.05. The Mantel analysis revealed an r value of 0.2–0.4 for the correlation between TP and the Simpson index. However, no significant correlation was found between the Shannon index, chao1 index, ACE index and any other variables (Mantel’s *p* ≥ 0.05).

This study employed a correlation network to further investigate the association between various environmental factors and the structure of soil fungal communities at the genus level. The results of the correlation network analysis were subjected to relationship deduplication and data screening (Figure 6). The analysis revealed that 37 species of soil fungi at the genus level exhibited associations with the measured environmental factors. Notably, AN, OM and AP displayed the strongest correlations with species, positively correlating with 30, 27 and 12 species, respectively. Conversely, pH exhibited the highest negative correlation with species, negatively correlating with five species. The genera Mortierella, Typhula, Wickerhamomyces, Trichosporon and Candida exhibited the highest correlation with environmental factors. Specifically, Mortierella displayed negative correlations with pH and TS, while exhibiting positive correlations with AP, OM and TK. Typhula, on the other hand, exhibited positive correlations with AP, TP, OM, AK, AN and TN. The presence of Wickerhamomyces exhibited a negative correlation with the levels of AP, TK, OM and TN and a positive correlation with pH. On the other hand, Trichosporon and Candida showed a negative correlation with AP, TK, OM and AN and a positive correlation with pH.

## 3. Discussion

### 3.1. Effects of Different Straw Returning on Soil Physicochemical Indices

Saline–alkali soils exhibit low levels of organic carbon, limited nutrient and water availability and inadequate soil structure, resulting in detrimental effects on crop productivity [29]. Research has demonstrated that the utilization of cost-effective and environmentally friendly organic materials can significantly enhance the characteristics of saline–alkali soils [30,31,32]. Previous research has demonstrated that the practice of straw returning has the potential to decrease soil pH and total salinity content [33,34]. The findings of this study provide additional evidence supporting this assertion. This phenomenon can be attributed to the fact that when straw is returned to the field, it provides a protective layer on the soil surface, thereby reducing evaporation and impeding the upward movement of salinity with water. Consequently, this process leads to a reduction in both soil pH and total salinity levels. Simultaneously, the reintroduction of straw into the field alters the soil structure, enhancing soil infiltration. Consequently, the downward movement of salt in the soil occurs through precipitation leaching and surface soil infiltration, thereby mitigating soil salinization [35]. Straw, being an organic material, undergoes decomposition upon being reintroduced to the field. This decomposition process facilitates the transfer of significant quantities of organic carbon and nitrogen substances to the soil. Additionally, it results in the production of organic matter, including polysaccharides, proteins and lignin. These organic compounds contribute to enhanced soil aeration, improved soil microecological conditions, increased microbial activity and the stimulation of crop roots to secrete or release additional organic compounds [36]. Crop straw is rich in several essential components, including carbon, nitrogen, phosphorus and potassium, so it serves as a valuable reservoir of soil nutrients. The enhancement of straw can lead to an increase in the levels of phosphorus, potassium, organic matter, total nitrogen and accessible nitrogen in the soil. Nitrogen is a crucial ingredient for the proliferation and maturation of agricultural plants [37,38]. Zhang et al. (2016) discovered that the practice of rice straw returning had a notable impact on the soil’s nitrogen content, particularly within the 0–20 cm soil layer [39]. The application of wheat straw in paddy fields resulted in a notable increase in soil conductivity, total nitrogen, carbon and nitrogen, and dissolved organic carbon content. This increase was directly related to the quantity of wheat straw applied [40]. The findings of this study indicate that the levels of alkali-hydrolyzable nitrogen and total nitrogen in the soil were elevated by the practice of incorporating wheat and rape straw back into the field. Furthermore, it was observed that the concentration of alkali-hydrolyzable nitrogen and total nitrogen in the soil layer was notably higher within the 10–20 cm depth range. In a study conducted by Mu et al. [41], it was demonstrated that the implementation of continuous straw returns, in conjunction with shallow soil tillage, resulted in a considerable augmentation of soil organic matter and accessible potassium levels within the 0–20 cm soil layer. The practice of straw returning has been found to have a positive impact on soil nutrient levels in paddy fields. Specifically, it has been observed to greatly enhance the availability of phosphorus in the soil, which, in turn, is absorbed and utilized by crops [42]. In a study conducted by Xu et al. [43], it was shown that the organic matter content and available potassium levels exhibited a rise of 3.8 g/kg and 1.0 mg/kg, respectively. However, no significant alteration was observed in the available phosphorus content. The findings of this study indicate that the levels of organic matter, available phosphorus and available potassium in wheat and rape straw were significantly greater compared to saline–alkali soil. However, the levels of total phosphorus and total potassium exhibited an increase but were not statistically significant. The usage of saline–alkali land holds significant importance in meeting China’s grain demand and promoting sustainable agricultural development, as it serves as a valuable reserve of farmed-land resources. The fundamental objective of improving and utilizing saline–alkali land is to employ diverse technologies to mitigate the adverse effects of soil salinization, enhance soil fertility and optimize the soil conditions to facilitate the optimal growth and development of crops. This approach can achieve the rational utilization of saline–alkali land resources [44]. Thus, the different straw return and return methods for saline soils deserve further research.

### 3.2. Effects of Different Straw Returning on Soil Fungal Community Structure and Diversity

Soil microorganisms have a vital role in the process of straw breakdown, as indicated by previous research [45]. Soil microbes are known to have significant contributions to several ecological processes, such as straw decomposition, nutrient cycling, plant growth and development, and soil fertility [46,47,48,49,50]. Furthermore, the diversity and composition of soil microbial communities are closely associated with the soil environment [51]. This study detected 724, 4567 and 5413 OTUs in the SCK, SXM and SYC groups, respectively. The ACE and Simpson indices exhibited considerably higher values in the SCK group compared to the SXM and SYC groups. Additionally, the Shannon indices for the SCK, SXM and SYC groups were determined to be 4.39, 7.31 and 8.13, respectively. Previous studies have proposed that the practice of returning sugarcane straw [52], wheat straw [53] and corn straw [54] to the soil has a notable impact on enhancing the richness and diversity of soil microbial communities. Specifically, the decomposition of rice straw has been found to significantly increase the diversity of soil fungal communities, while the diversity of bacterial communities remains unaffected [55]. These findings indicate that the process of straw decomposition introduces certain bacteria and fungi from the straw into the soil, thereby augmenting the diversity and uniformity index of microorganisms in the soil [56]. The results of this study align with the aforementioned findings. The soil fungal communities in SXM and SYC exhibited greater richness and diversity compared to those in SCK. Furthermore, the increase in richness and diversity was more pronounced in SYC. Additionally, the application of straw returning to the field was found to enhance the diversity of microbial communities [57,58]. Generally, an increase in the diversity of soil microbial communities is associated with a more intricate interplay between microorganisms and the soil environment, resulting in enhanced stability of soil ecosystems and heightened resistance against diseases [59,60,61]. This suggests that the utilization of SXM and SYC can potentially contribute to the enhancement of soil function, ecosystem stability and disease resistance. Within the fungal community, the prevailing fungi observed in SXM and SYC were *Mortierella*, *Plectosphaerella*, *Ascomycota*, *Fusarium* and *Alternaria*. It was noted that the presence of wheat straw and corn straw in the field led to an increase in the relative abundance of *Chaetomium* and *Fusarium* [23]. Additionally, the application of straw to sandy loam and medium loam soil resulted in an increased relative abundance of *Mortierella*. The prevalence of fungal communities in *Alternaria* and *Ascomycota*, as indicated by previous research [62,63], aligns with the findings of our study. In their investigation, Guo et al. (2022) examined the impact of straw incorporation on soil physicochemical characteristics and the diversity of fungal community structure. The study revealed that the practice of straw returning resulted in an elevated proportion of *Ascomycota* and *Mortierella* species [53]. In terms of fungal function prediction, straw application increased symbiotroph and pathotroph relative abundance; however, increased pathotroph abundance is negative for soil health, which may be related to the texture of saline soils. Yu et al. 2022 suggested that legume cover crops can suppress pathotrophic fungal microorganisms [2,64], which could inform further saline improvement trials later. We concluded that the implementation of straw returning resulted in a notable enhancement in the abundance and diversity of soil fungus, as compared to the control group. Enriched fungal communities contribute to saline soil health and functioning. Moreover, increased Ascomycota abundance promoted organic matter degradation. But the mechanism of different types of straw returning to the field on the abundance of different fungal communities needs to be further investigated to explore the effect of straw on specific fungal communities.

### 3.3. Environmental Drivers of Soil Fungal Communities

The findings of this study indicate that AN, OM and AP exhibited the strongest correlations with species. Specifically, AN was positively correlated with 30 species, while OM and AP were positively correlated with 27 and 12 species, respectively. Conversely, AN, OM and AP were negatively correlated with two, four and two species, respectively. These results suggest that OM, AN and AP play significant roles in shaping the structure of soil fungal communities in saline–alkali soil. Studies have indicated that soil pH plays a significant role in influencing the viability and reproductive capabilities of soil microorganisms. Additionally, it has the potential to disrupt the structure of soil microbial communities by impacting the composition, chemical characteristics and utilization efficiency of soil substrates [65]. The observed inverse relationship between pH levels and total nitrogen (TN) content, as well as microbial communities, aligns with previous investigations [66,67]. Consequently, the makeup of soil microbial communities is impacted [68], thereby highlighting the significance of organic matter as a key determinant of fungal communities [69]. The findings of AN and AP in a study conducted by Sun et al. [70] exhibited a resemblance to the results observed in the present investigation. The findings from the analysis of community and environmental factor correlation networks revealed that AN exhibited a positive correlation with 30 species, including *Archaeorhizomyces*, *Strelitziana*, *Pichia*, *Candida*, *Trichosporon* and *Wickerhamomyces*. Similarly, OM displayed a positive correlation with 27 species, which were largely consistent with those associated with AN. On the other hand, AP exhibited a positive correlation with 12 species, such as *Didymellaceae*, *Rozellomycota*, *Tricholoma*, *Thermoascus*, *Leotiomycetes* and *Trichomerium*. In contrast, pH exhibited a negative correlation with five species, namely *Archaeorhizomyces*, *Strelitziana*, *Candida*, *Trichosporon* and *Wickerhamomyces*. Notably, only a limited number of species demonstrated a positive correlation with pH. These findings suggest that environmental factors, like OM, AN and AP, can facilitate the growth and reproduction of diverse fungal communities, while pH acts as an inhibitory factor for the majority of microbial communities. Alterations in the composition of nitrogen (N), phosphorus (P) and potassium (K) are crucial elements that are necessary for the development and reproductive processes of soil microorganisms [71]. In conclusion, the impact of environmental factors on microbial community structure is complex, and it can lead to alterations in the composition and development of microbial communities through both direct and indirect mechanisms. Therefore, enhancing the soil environment and enhancing the diversity of the soil fungal community can facilitate the recycling of soil nutrients and enhance the overall quality of saline–alkali soil.

## 4. Materials and Methods

### 4.1. Overview of the Test Site

The testing facility is situated within the rape test base of Huangcitan (36.7181° N, 103.6348° E), specifically in Shangchuan Town, Yongdeng County, Lanzhou City, Gansu Province. The site experiences a yearly average temperature range of 1 to 15 °C, encompassing both high and low temperatures. The recorded minimum temperature was −23 °C. The maximum temperature recorded in the given year was 33 °C, while the annual precipitation amounted to 696.9 mm. The soil composition of the area is predominantly sandy, characterized by high levels of salinity and alkalinity. Furthermore, the soil exhibits low fertility and limited agricultural potential. The typical soil physicochemical properties include: a pH value of 8.91, a total salt content of 0.71%, an organic matter content of 5.38 g∙kg^−1^, an alkali-hydrolyzable nitrogen content of 43.51 mg∙kg^−1^, an available phosphorus content of 2.95 mg∙kg^−1^, an available potassium content of 153.33 mg∙kg^−1^, a slow-acting potassium content of 743.78 mg∙kg^−1^, a total nitrogen content of 0.56 g∙kg^−1^, a total phosphorus content of 0.61 g∙kg^−1^ and a total potassium content of 19.69 g∙kg^−1^.

### 4.2. Experimental Design

The experiment commenced in July 2021 and concluded in June 2023. It consisted of three distinct treatments: a control group representing saline–alkali soil (SCK), a treatment involving the return of wheat straw (SXM) and a treatment involving the return of rapeseed straw (SYC). A randomized block test was used with nine (30 m^2^, per plot) experimental plots with three replications. The wheat variety used in the study was “Longchun 10”, and the oilseed rape variety was “Longyuo 7”, which were planted in conventional farm land. We harvested the above-ground parts of oilseed rape and wheat at the early flowering stage, crushed them to small sections of 3–5 cm and spread the straw uniformly on the surface of the saline field at a dosage of 4 kg/m^2^, and buried them into the soil at a depth of 0–30 cm by ploughing. After 2 years of conventional field management measures, timely and appropriate amount of watering to keep the field soil moist, the straw was decomposed by microorganisms in the soil or directly into the soil.

### 4.3. Soil Sampling and Analysis

In mid-July 2023, soil samples were obtained and subsequently categorized into three distinct soil layers: soil layer 1 (0–10 cm), soil layer 2 (10–20 cm) and soil layer 3 (20–30 cm). We utilized the diagonal 5-point sampling technique. The soil sample was carefully sieved and promptly transferred to a refrigerated container, ensuring its preservation during transportation to the laboratory. Upon arrival, the soil was partitioned into two distinct portions. One portion was stored in a freezer set at −20 °C to facilitate efficient analysis of microbial content. The other portion of the soil sample was subjected to natural air-drying, enabling subsequent assessment of its physical and chemical characteristics.

Conventional methods are employed to assess the physical and chemical characteristics of soil. The soil pH value (water: soil, 2.5:1) was calculated using the pHS-25 acidity meter (INESA Scientific InstrumentCo., Ltd., Shanghai, China) via the electrode method [72]. The total soil salinity (TS) was determined using the conductivity method [73]. The determination of organic matter (OM) was conducted using the potassium dichromate sulfuric acid oxidation external heating method [74], employing the DDS-12A digital display conductivity meter (Shanghai Hongyi Instrumentation Co., Shanghai, China). The alkali-hydrolyzable nitrogen (AN) was determined using the alkaline hydrolysis and diffusion method [75], utilizing the 28VX-500 electrothermal constant temperature incubator (Zhengzhou Likang Instrument Equipment Co., Zhengzhou, China). The Cary50 ultraviolet-visible spectrophotometer (Mettler Toledo International Ltd., Greifensee, Switzerland) was employed for the analysis. This study employed the sodium bicarbonate extraction-molybdenum-antimony anti-colorimetric method [76] to determine the available phosphorus (AP). The determination of available potassium (AK) was carried out using the Sherwood M410 flame photometer (Shanghai Shujun Instrument Co., Shanghai, China) with ammonium acetate extraction-flame photometry [77]. The Sherwood M410 flame photometer and Kicttev8200 Kjeldahl nitrogen determination instrument were utilized for the determination of total nitrogen (TN) through the Kjeldahl method [78]. The Cary50 UV-Vis spectrophotometer was employed to determine total phosphorus (TP) using the alkali melting-molybdenum-antimony anti-colorimetric method [79]. Lastly, total potassium (TK) was determined by alkali melt-flame photometry using the Sherwood M410 flame photometric juice [80].

### 4.4. Soil DNA Extraction, Illumina Sequencing

Soil DNA was isolated from a 0.5 g sample of fresh soil using the OMEGA DNA kit (Biomarker Technologies, Beijing, China), following the provided instructions. The amplification conditions were (95 °C for 5 min, 95 °C for 1 min, 50 °C for 30 s, 72 °C for 1 min, 72 °C for 7 min, 4 °C for ∞) for five cycles. The ITS1 (endo-transcriptional region 1) of the fungal rRNA gene was then amplified from the collected DNA, primarily focusing on the ITS region. Fungal primers listed in Table 2 were utilized for this purpose [81]. The amplification process was conducted utilizing a Long Gene^®^A300 thermal cycler (Hangzhou BORI Technology Co., Hangzhou, China). To obtain the PCR product, three replicate soil samples were combined throughout the amplification procedure. The concentration of PCR products was quantified using the DeNovix DS-11 instrument. Subsequently, a small fragment library was prepared for sequencing on the Illumina NovaSeq platform using the paired-end sequencing method. The library was then sent to Beijing Biomarker Biotechnology Co., Ltd. (Beijing, China) for sequencing.

### 4.5. Statistical Analysis

The raw readings received from sequencing were filtered using Trimmomatic v0.33 software as the initial step. The primer sequences were identified and removed using the cutadapt 1.9.1 software. The resulting reads without primer sequences were considered as clean reads. Sequences were denoised for OTUs using the dada2 method [82] in the QIIME2 (versoin 2020.6, https://qiime2.org/, accessed on 10 September 2023) software [83]. The duplex reads were spliced and removed using USEARCH (version 10) from chimeras (UCHIME, version 8.1) to yield high-quality sequences for subsequent analysis. The database is UNITE (Release 8.0, https://unite.ut.ee/, accessed on 26 September 2023). The alpha diversity index of the samples was assessed at various sequencing depths using QIIME2 and R language tools, taking into account the sequencing volume of each sample. The resulting data were visualized using imageGP (https://www.bic.ac.cn/BIC/#/, accessed on 22 October 2023) online software. PCA analysis was conducted on each sample using the Bray–Curtis calculation method [84], based on the feature-unweighted distance. The species composition stacks were mapped using Chiplot (https://www.chiplot.online, accessed on 7 October 2023) online software. The fungi were classified into three categories, namely pathotroph, symbiotroph and saprotroph, based on their nutritional mode. This classification was determined using the FUNGGuild (Fungi Functional Guild) tool, which relied on information from the published literature or authoritative websites. Additionally, functional gene prediction of fungal colonies was performed [85]. The data were screened using Spearman’s rank correlation analysis, specifically selecting correlations greater than 0.1 and *p*-values less than 0.05. This analysis was conducted in the R language to construct a species correlation network graph. The network graph included various node properties, such as degree, clustering coefficient, compact centrality, intermediary centrality, Zi (within-module connectivity) and Pi (among-module connectivity) [86]. Furthermore, it is worth noting that nodes can be classified into four distinct categories based on Zi and Pi values, as proposed by Zi and Pi. These categories include peripheral nodes, connectors, module hubs and network hubs [82]. Peripheral nodes are characterized by low values of zi (≤2.5) and Pi (≤0.62), possessing only a limited number of edges and typically being connected solely to nodes within their respective module. Connectors, on the other hand, exhibit low values of zi (≤2.5) but higher values of Pi (>0.62), often establishing connections between different modules. Module hubs, characterized by high values of zi (>2.5) and low values of Pi (≤0.62), are extensively connected to numerous nodes within their module. Lastly, network hubs, with high values of zi (>2.5) and Pi (>0.62) [82], display a high degree of connectivity, both within their module and across different modules. Initially, the abundance data of the species samples were utilized for conducting a detrended correspondence analysis (DCA). Subsequently, the R language vegan package was employed to perform redundancy analysis (RDA) and generate a plot depicting the combination of a heatmap and network. This plot was based on the correlation between environmental factors and species, as well as the Alpha index (Pearson) and Mantel analysis. The statistical software SPSS 27 was utilized to conduct a significant differences test, with a predetermined significance level of *p* < 0.05.

## 5. Conclusions

Following a two-year experiment aimed at improving saline–alkali land, wheat and rape straw were returned to the field after undergoing a series of physical and chemical alterations in the soil. This intervention effectively enhanced the overall fertility of the saline–alkali soil, particularly by significantly increasing the levels of AN, AP, OM and TN in the rape straw. Notably, the variation between different soil layers was not statistically significant. Additionally, the soil pH and TS contents were effectively reduced as a result of the intervention. At the taxonomic level of the genus, there was an observed in-crease in the relative abundance of *Mortierella*, *Ascomycota*, *Fusarium* and *Alternari*, while the relative abundance of *Enterocarpus* exhibited a substantial decrease. The composition of soil fungal community structure on saline–alkali land was primarily influenced by OM, AN and AP. These environmental factors played a significant role in shaping the distribution patterns of soil fungi, including *Mortierella*, *Typhula*, *Wick-erhamomyces*, *Trichosporon* and *Candida*.

## Figures and Tables

**Figure 1 microorganisms-12-00277-f001:**
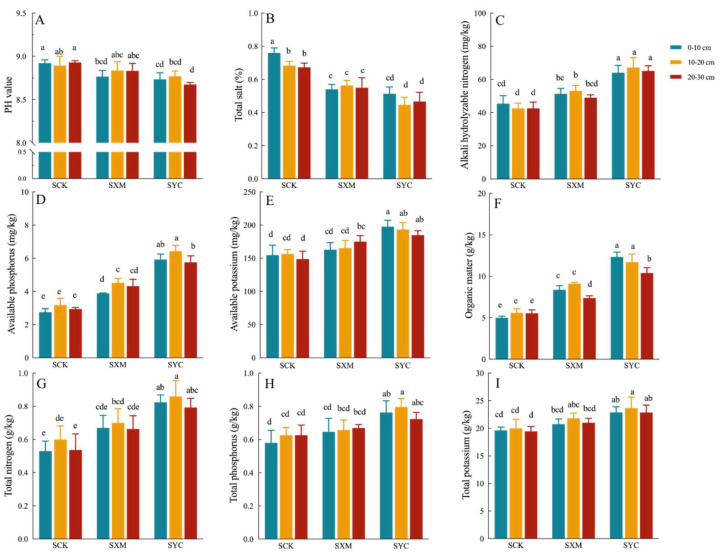
The changes in pH, total salt content and soil nutrients in saline–alkali soil under different straw-returning treatments. (**A**) PH. (**B**) Total salt. (**C**) Alkali hydrolyzable nitrogen. (**D**) Available phosphorus. (**E**) Available potassium. (**F**) Organic matter. (**G**) Total nitrogen. (**H**) Total phosphorus. (**I**) Total potassium. Lowercase letters indicate the significance between different soil layers and treatments (*p* ≤ 0.05).

**Figure 2 microorganisms-12-00277-f002:**
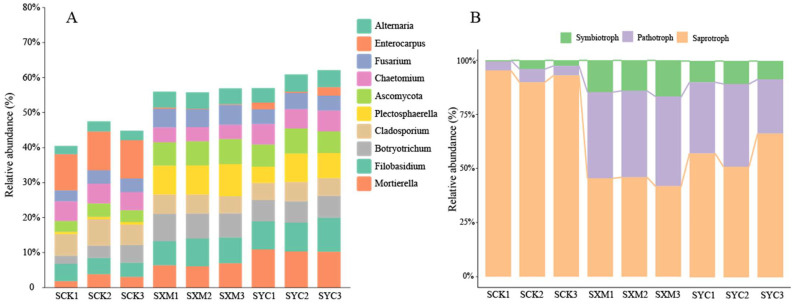
Prediction of soil fungal community composition and function under different treatments. (**A**) Species composition tacked plot, genus level, the top-10 species. (**B**) FunGuild fungal function analysis chart.

**Figure 3 microorganisms-12-00277-f003:**
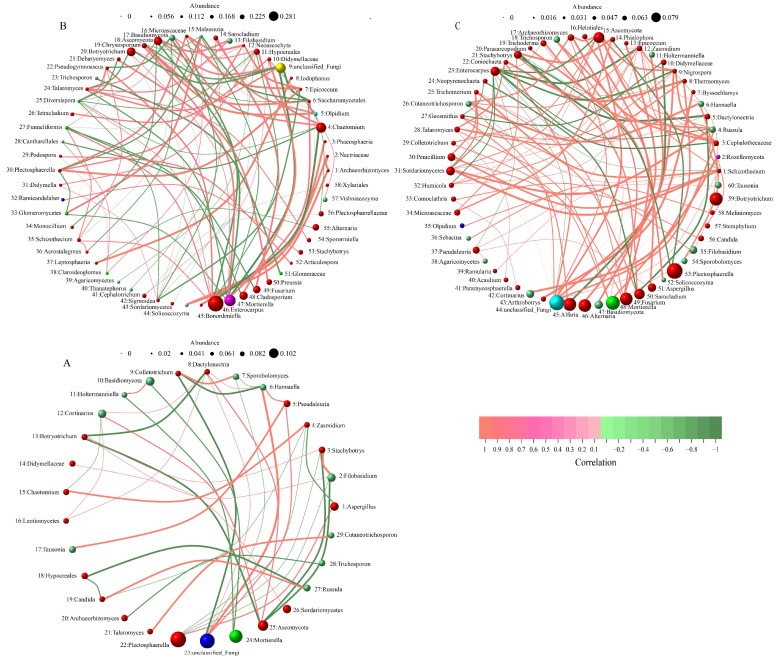
Network diagram of species at the level of genera, (**A**) SCK, (**B**) SXM, (**C**) SYC, genus level, species number selection: 70, correlation type: Spearman, correlation coefficient threshold: 0.1, correlation *p*-value threshold: 0.05.

**Figure 4 microorganisms-12-00277-f004:**
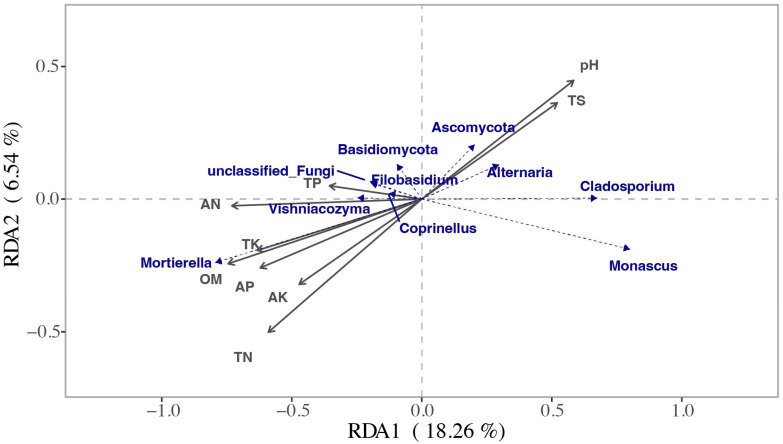
Redundancy analysis of species and environmental factors of soil fungi in different treatments (RDA), belonging to the level, the top 10-species.

**Figure 5 microorganisms-12-00277-f005:**
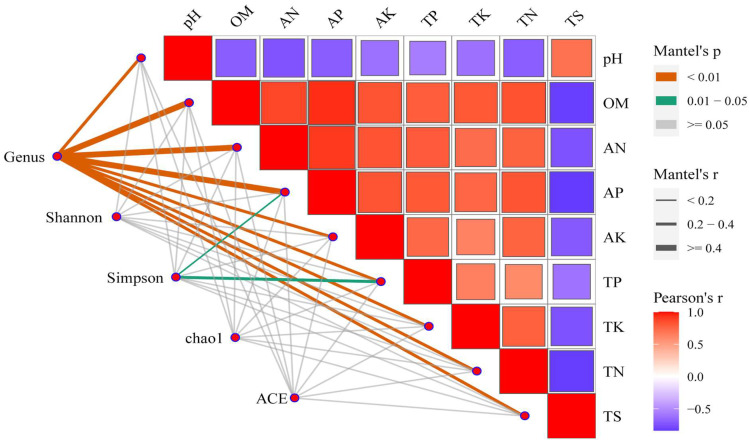
Heatmap and network combination map of species and environmental factors and Alpha correlation. (1) Illustration: Mantel’s *p* is the *p*-value of Mantel analysis of environmental factors and species and Alpha index; Mantel’s r is the r-value of Mantel analysis of environmental factors and species and Alpha index; Pearson’s r is the correlation *p*-value of environmental factors and species and Alpha index. (2) Heatmap in the upper-right corner: there is a correlation between environmental factors and environmental factors, the color red and blue of the heatmap indicate positive correlation and negative correlation respectively, and the size of the heat map block is consistent with the correlation r; (3) the network diagram of the lower-left corner: the network relationship between species, Alpha index and environmental factors, the color of the line is consistent with the Mantel’s *p* in the legend, and the thickness of the line is the same as that in the legend Mantel’s r.

**Figure 6 microorganisms-12-00277-f006:**
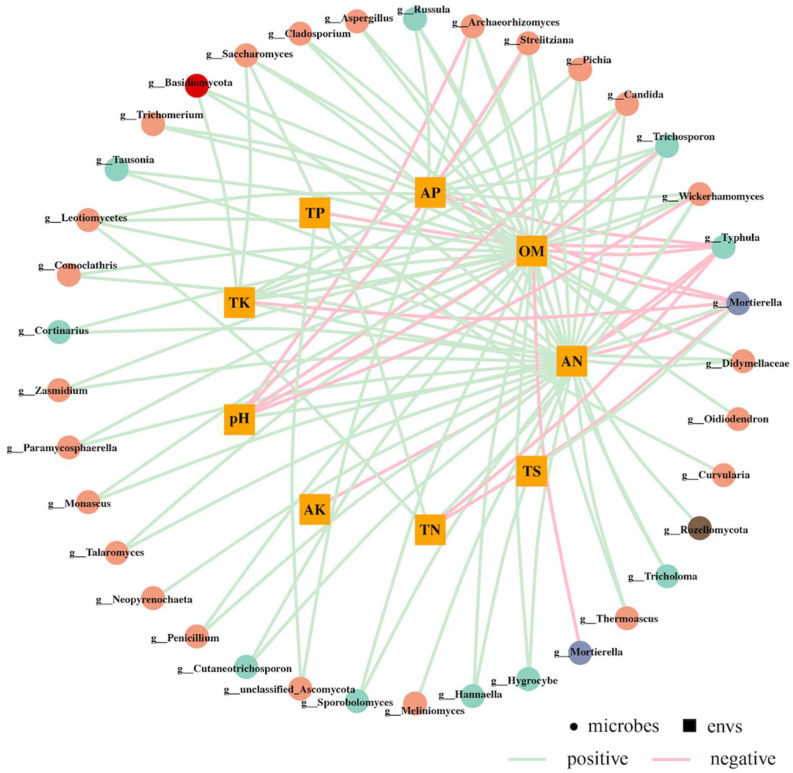
Correlation network diagram between species and environmental factors, genus level, Pearson correlation, correlation threshold 0.3, correlation *p*-value threshold 0.05, *p*-value node number 80, edge number 100.

**Table 1 microorganisms-12-00277-t001:** Environmental factors explain the importance order and significance of test results of soil fungi in different treatments.

Environment Factors	Order of Importance	Explains/%	*p*-Value
TN	1	47.74%	0.001
OM	2	47.21%	0.001
pH	3	42.78%	0.003
AN	4	41.66%	0.002
AP	5	34.93%	0.007
TK	6	33.60%	0.013
TS	7	31.68%	0.01
AK	8	25.55%	0.032
TP	9	10.28%	0.269

**Table 2 microorganisms-12-00277-t002:** Fungal parameter information.

Content	Fungal Parameter Information
Amplicon information	its its1_f
Primer information	F:5′-CTTGGTCATTTAGAGGAAGTAA-3′;
	R:5′-GCTGCGTTCTTCATCGATGC-3′
Species annotation methods	Bayesian
Species annotation database	UNITE
Feature acquisition method	dada2
Feature filters	2

## Data Availability

Data are contained within the article and Appendix A.

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
