# Peer review of "Impact of Straw Incorporation on the Physicochemical Profile and Fungal Ecology of Saline–Alkaline Soil"

_microorganisms, 2024, doi:10.3390/microorganisms12020277_

Round 1

Reviewer 1 Report

Comments and Suggestions for Authors

Please see attached files for specific comments 

Comments on the Quality of English Language

Overall authors are too verbose, sometimes leading to loss of nuances within each section. The english grammar and typos throughout the text need improvement, Foe specific comments see attached file.

Reviewer 2 Report

Comments and Suggestions for Authors

The authors propose a fungal microbial ecology study for analyzing the beneficial effect of straw treatment on agricultural soil.

I think that the experimental design is correct, the time longitude of the study is appropriate, the number of treatments/replicates is adequate.

The methods applied seem appropriate, although there is room for improvement.

I have some suggestions that I encourage the authors to consider:

1) I do not really like the wording on the subsection 4.5.

There are some mistakes in the writing that I would like the authors to adjust.

As an example, the authors stated that they used DADA2 algorithm over the OTUs to denoise. This is wrongly written. DADA2 algorithm works over the sequences of the samples, not the OTUs. OTUs are computational constructs based on clusterization of sequences over some similarity  criteria. The authors should re-write this sentence.

I would like the authors to specifically write down what software they used for chimera detection, and if the chimera detection was done using a database (which is very uncommon) or de novo (which is the usual).

I would like the authors to specifically write down the version of the UNITE database used for the taxonomic classification. I think that this is important because UNITE database is updated frequently.

The authors stated that denoised, non-chimeric sequences were then clustered into OTUs using mothur. I think that using both DADA2 denoising and OTU clusterization is a very good approach. However, I would like the authors to specifically write down the clusterization algorithm used for OTU clusterization in mothur and the similarity threshold used for clustering.

I would like the authors to specifically write down the alpha-diversity indices computed.

For beta-diversity analyses, did the authors use compositional statistics? Keep in mind that microbiome data has compositional nature and should be worked as such. I believe the authors did not use this approach because they stated that they used Bray-Curtis distance for PCA computation. In such a case, please cite a previous research in which the PCA computation was done in the same way the authors did.

2) I did not see any statistical analysis in which the authors try to discern fungal phylotypes that are characteristic for a particular treatment. I believe that this could be a very good addition to the manuscript, as the author might observe potential beneficial and/or problematic fungal phylotypes appearing only in one of the treatments, which would add valuable information over its implications for a potential real use. At the same time, dominant fungal phylotypes that are not characteristic of any particular treatment could be core fungal microbiome, which might also be important to define. I strongly recommend the authors to develop such analysis, as it will add even more material for them to discuss.

Author Response

Response to Reviewer 2 Comments

Dear Reviewer,

We are absolutely thrilled to express our heartfelt gratitude for the time you have taken to provide us with your constructive remarks and invaluable suggestions. Your feedback has been instrumental in significantly elevating the quality of our manuscript and has enabled us to make substantial improvements. We have taken every single one of your suggested revisions and comments into careful consideration and have incorporated them with utmost accuracy. Please find below our point-by-point response to your comments and revisions.

Reviewer 2

Comment: 1. The authors stated that they used DADA2 algorithm over the OTUs to denoise. This is wrongly written. DADA2 algorithm works over the sequences of the samples, not the OTUs. OTUs are computational constructs based on clusterization of sequences over some similarity criteria. The authors should re-write this sentence.

Response: Thank you for your comment, we have made the correction to "Sequences were denoised using the dada2 method to obtain OTUs".

Comment: 2. I would like the authors to specifically write down what software they used for chimera detection, and if the chimera detection was done using a database (which is very uncommon) or de novo (which is the usual).

Response: Thank you for your comment, Use of de novo, we used (USEARCH, version 10) to splice double-ended reads and remove chimeras (UCHIME, version 8.1), resulting in high-quality sequences for subsequent analysis.

Comment: 3. I would like the authors to specifically write down the version of the UNITE database used for the taxonomic classification. I think that this is important because UNITE database is updated frequently.

Response: Based on your comments. We filled in the database version as UNITE (Release 8.0, https://unite.ut.ee/).

Comment: 4. The authors stated that denoised, non-chimeric sequences were then clustered into OTUs using mothur. I think that using both DADA2 denoising and OTU clusterization is a very good approach. However, I would like the authors to specifically write down the clusterization algorithm used for OTU clusterization in mothur and the similarity threshold used for clustering.

Response: Thank you so much for your comment, I'm sorry it was a writing mistake on our part, we used the both DADA2 denoising and OTU clusterization. We've corrected our mistake.

Comment: 5. I would like the authors to specifically write down the alpha-diversity indices computed.

Response: Thank you for your comment, we have made additions based on your suggestions:"The alpha diversity index of the samples was assessed at various sequencing depths using QIIME2 (https://qiime2.org/) and R language tools".

Comment: 6. For beta-diversity analyses, did the authors use compositional statistics? Keep in mind that microbiome data has compositional nature and should be worked as such. I believe the authors did not use this approach because they stated that they used Bray-Curtis distance for PCA computation. In such a case, please cite a previous research in which the PCA computation was done in the same way the authors did.

Response: Thank you for your comment, we filled in the reference for this approach is "[83]".

Comment: 7. I did not see any statistical analysis in which the authors try to discern fungal phylotypes that are characteristic for a particular treatment. I believe that this could be a very good addition to the manuscript, as the author might observe potential beneficial and/or problematic fungal phylotypes appearing only in one of the treatments, which would add valuable information over its implications for a potential real use. At the same time, dominant fungal phylotypes that are not characteristic of any particular treatment could be core fungal microbiome, which might also be important to define. I strongly recommend the authors to develop such analysis, as it will add even more material for them to discuss.

Response: Thank you for your comment, and based on your comments we have added PERMANOVA analysis (Figure S3B). A statistical method for analysing similarity between groups of multidimensional data. Are the differences between sample points in the analysis for different subgroups significant。

Reviewer 3 Report

Comments and Suggestions for Authors

Comments on methodology:

1. Straw returning practice is important for maintaning soil quality and fertility and it is used by farmers all over the world. In this article wheat and rape straw was applied into a saline-alkali soil in a 2-year plot exteriment. However, the desription of the Experimental design in 4.2 is far from satisfactory. More information is needed with respect to:

- area of plots,

- amounts of straw applied per plot or ha,

- crop rotation; or where wheat and rape grown in 2-year monocultures?,

- when (after harvest?) and how many times was straw applied,

- how deep was straw incorporated into the soil?,

- „The straw was managed through crushing, burying it in the soil by tillage and plowing, and subsequently reintroducing it to the field following a sequence of physical and chemical transformations [73]” – this sentence is very unclear! What does it mean ...following a sequence of physical and chemical transformations … ? Where were these transformations conducted?

2. Positions of literature should be given for each soil parameter measured in this work.

Comments on results:

This article deals with soil fungal communities, however in lines 213-219 bacteria are discussed instead of fungi? Moreover, Figure 2B indicates that communitues of pathotrophic fungi inceresed in SYC and SXM treatments, which is ruther a negative outcome, but this problem is omitted in Discussion!

Author Response

Response to Reviewer 3 Comments

Dear Reviewer,

The authors would thank the editor and the reviewers very much for kindly spending such a time to provide relevant comments that are very helpful to us in the improvement of the manuscript. The authors’ responses to the reviewers’ comments are as the following:

Reviewer 3

Comment: 1. Straw returning practice is important for maintaning soil quality and fertility and it is used by farmers all over the world. In this article wheat and rape straw was applied into a saline-alkali soil in a 2-year plot exteriment. However, the desription of the Experimental design in 4.2 is far from satisfactory. More information is needed with respect to:

  • - area of plots,
  • - amounts of straw applied per plot or ha,
  • - crop rotation; or where wheat and rape grown in 2-year monocultures?,
  • - when (after harvest?) and how many times was straw applied,
  • - how deep was straw incorporated into the soil?,
  • - „The straw was managed through crushing, burying it in the soil by tillage and plowing, and subsequently reintroducing it to the field following a sequence of physical and chemical transformations [73]” – this sentence is very unclear! What does it mean ...following a sequence of physical and chemical transformations … ? Where were these transformations conducted?

Response: Many thank you for your valuable comment. For comments 1) - 5) we make the following changes: “A randomised block test was used with nine (30 m2, per plot) experimental plots with three replications. The wheat variety used in the study was "Longchun 10", and the oilseed rape variety was "Longyuo 7", which were planted in conventional farmland, harvested the above-ground parts of oilseed rape and wheat at the early flowering stage, crushed them to small sections of 3-5 cm, and spread the straw uniformly on the surface of the saline field at a dosage of 4 kg/m2 , and buried them into the soil at a depth of 0-30 cm by ploughing and ploughing.”

For comment 6) we replaced another formulation: “After 2 years of conventional field management measures, timely and appropriate amount of watering to keep the field soil moist, the straw was decomposed by microorganisms in the soil or directly into the soil.”

Comment: 2. Positions of literature should be given for each soil parameter measured in this work.

Response: Thanks to your comments, we filled in the appropriate literature. "[71-79]".

Comment: 3. Comments on results: This article deals with soil fungal communities, however in lines 213-219 bacteria are discussed instead of fungi? Moreover, Figure 2B indicates that communitues of pathotrophic fungi inceresed in SYC and SXM treatments, which is ruther a negative outcome, but this problem is omitted in Discussion!

Response: Many thank you for your valuable comment. Upon inspection, this was a writing error on our part and we described fungi. We have revised it according to your comments. In addition, we add to our discussion of pathotrophic fungal communities.
